# The Coupling Coordination Measurement, Spatio-Temporal Differentiation and Driving Mechanism of Urban and Rural Water Poverty in Northwest China

**DOI:** 10.3390/ijerph20032043

**Published:** 2023-01-22

**Authors:** Yun Ding, Shiqi Zhang, Ruifan Xu, Yuan Gao, Hao Ding, Pengfei Sun, Wenxin Liu

**Affiliations:** 1School of Economics and Management, Northwest A&F University, Yangling 712100, China; 2College of Forestry, Henan Agricultural University, Zhengzhou 450002, China; 3School of Economics and Management, Shandong Agriculture University, Taian 271018, China

**Keywords:** urban and rural water poverty, coupling coordination, spatio-temporal differentiation, driving mechanism

## Abstract

Regarding the background of the “urban–rural dual structure”, the scientific evaluation of the relationship between urban and rural water resource systems is of great significance for alleviating water use contradictions and optimizing water resource allocation. Based on the theory of water poverty, the coupling coordination model was used to quantify the relationship between the urban and rural water resource systems in northwest China from 2000 to 2020; furthermore, the spatial and temporal evolution characteristics and driving mechanism were studied by using spatial autocorrelation, a hot spot analysis and the Tobit model. The result showed the following: ① The scores of urban and rural water poverty have risen significantly, and the urban and rural water resource systems have improved significantly. Among them, urban water poverty demonstrated a tiered pattern of “east–middle–west”, and rural water poverty demonstrated a pattern of collapse of “high on both sides and low in the middle”. ② The overall degree of coupling coordination between urban and rural water poverty has greatly improved. However, nearly 70% of the regions are still of the basic uncoordinated type, and the differences between regions have been gradually expanding, showing a state of agglomeration in space, mainly of the low–high and high–high agglomeration types. The hot spot area was mainly concentrated in the southeast area, showing a gradual expansion trend, and the cold spot area was mainly concentrated in the central area, showing a gradual shrinking trend. ③ The level of economic development, industrial structure and agricultural production demonstrated a positive impact on the degree of coupling coordination. The degree of industrialization, the level of opening, technological progress, population size, expenditure on supporting agriculture and environmental regulation had different effects on the degree of regional coupling coordination. Different strategies should be adopted to promote the coupled and coordinated development of urban and rural water resource systems.

## 1. Introduction

Ensuring that all people have “clean drinking water and sanitation facilities by 2030” is one of the 17 sustainable development goals (SDGs) set by the United Nations. However, nearly 4 billion people in the world face serious water shortages for at least one month in a year, more than 3 billion people lack sufficient cleaning facilities and 80% of the wastewater is discharged into the environment without treatment, which seriously threatens human health [1]. As a traditional arid region in China, northwest China has unique natural resources and socio-economic characteristics. First of all, the lack of water resources is the main factor limiting human social progress and economic development in the region. There are 32 regions in northwest China with per capita water resources less than 2000 m^3^, of which, 26 regions have per capita water resources less than the minimum of 1000 m^3^ and are in a state of water shortage or serious water shortage. At the same time, the overexploitation and extensive utilization of water resources not only affect the sustainable utilization of water resources, but also further restrict the sustainable social and economic development of the region [2]. Secondly, poverty is the main factor limiting the improvement in water resource management capacity in this region. The number of people in rural poverty in northwest China totals 920,000, accounting for 16.7% of the rural poverty in China. The average poverty incidence rate is 1.34%, which is much higher than the national poverty incidence rate of 0.6%. Backward rural infrastructure, continuous population reduction and a lack of cultural and human capital have become “stumbling blocks” to improving the water resource system [3,4]. Finally, the ecological environment in northwest China is fragile, and its carrying capacity is limited. This region includes three of China’s four ecological fragile zones–desert, loess and alpine regions–including three of the country’s five ecologically fragile provinces—Gansu, Ningxia and Qinghai. Although the ecological restoration project represented by returning farmland to forest and grassland has been gradually implemented, the regional conditions such as a high soil erosion rate, deep land desertification and desertification, and “lack of forest and little green” have not changed [5]. In a word, affected by large-scale and high-intensity human development and construction activities, various water resource problems such as the low efficiency of industrial and agricultural water use, mismatching of residents’ water supply and consumption, uneven construction of urban and rural water conservancy facilities and serious pollution of rural water bodies are intertwined, which seriously hinder the sustainable development of resources, the economy and the environment [6,7,8]. Facing the current situation of the development gap between urban and rural water resources increasing, the No. 1 central document in 2021 proposed to “establish a balanced allocation mechanism of urban and rural public resources, and strengthen overall planning and top-level design”. Therefore, it is urgent to scientifically evaluate the interaction mechanism of urban and rural water resource systems and explore the driving factors behind these systems, which are of great significance, in order to solve the dilemma of water resource shortages and pollution in northwest China and realize the goal of sustainable utilization of resources, sustainable social and economic development and the benign cycle of the ecological environment.

In recent years, water resource evaluation has been a dynamic process with expanding connotations and extensions. From water quantity evaluation, to water quality evaluation, and then to the multi-scale comprehensive evaluation of water resources, it has increasingly shown the characteristics of integration, synthesis and dynamics [9,10,11]. Caroline Sullivan, a Swedish hydrologist, proposed the water poverty index (WPI), which is regarded as an integrated analysis tool for analyzing water resource management issues. The index comprehensively considers five dimensions, including resources, access, capacity, use and the environment, so that people can have a deeper understanding of the relationship between the availability of water resources, water safety and human welfare [12]. Subsequently, Julie Wilk further improved the water poverty index and put forward the water prosperity index. In the process of index selection, relative attention was paid to the selection of indicators concerning facilities and capacity, which explained the important roles of space and gender in improving the situation of regional water resources [13]. Sullivan used the principal component method to analyze the shortage of water resources in MENA, taking into account the impact of politics, society, ecology and the economy on the allocation of water resources in the region [14]. Sun analyzed the coupling coordination relationship between water poverty and economic poverty, urbanization and industrialization [15,16]. Zhao analyzed the temporal and spatial pattern changes in the coupling coordination relationship between rural water poverty and agricultural modernization, which greatly enriched and promoted the theoretical and empirical research in this field [17]. In addition, spatial metrology software such as GIS and remote sensing have been increasingly applied to regional evaluation. By searching, storing, processing and analyzing geospatial data, they can exert unique visual effects on the base map, and improve the accuracy and reliability of water resource management policies [18]. Taking coastal cities in Africa as an example, Alioune Kane and Vincent Turmine used GIS to analyze the important role of facility and capacity subsystems in solving economic poverty and water resource shortages [19]. Zhang used a POME two-level fuzzy pattern recognition model to measure agricultural water poverty in China, and analyzed its spatial pattern dynamics from the perspective of space–time [20]. Kallio used the data-driven weight method to analyze the heterogeneity of water poverty in 8215 villages in Laos from the two dimensions of time and space, and determined that poverty, commercial or subsistence agriculture and village location were the three main drivers of water poverty [21]. The above research results have revealed the formation mechanism, and temporal and spatial characteristics of water poverty through the superposition of quantitative models; expanded the theoretical evaluation method of water poverty; and provided a good research foundation for this paper. Nevertheless, there are still many problems to be solved in the study of water poverty. Firstly, the index system of the water poverty measurement model is not reasonable. On the basis of pursuing comprehensiveness, a large number of indicators are used to evaluate the water resource system. The selection of indicators, which initially outlines the multi-dimensional characteristics of the water resource system, is one-sided and limited. It focuses more on the description of resource endowment, and it pays insufficient attention to human welfare or the social development level, which cannot comprehensively interpret the rich connotation of water resource sustainability. Based on the perspective of poverty concept, “promoting common prosperity through high-quality development” undoubtedly adds a new meaning to the connotation of poverty–human well-being. Secondly, there have been few studies on the interaction mechanism between urban and rural water resources. The urban–rural dual structure left over from China’s planned economy era determines the dominant role of cities in water resource utilization and solidifies the unbalanced pattern of water resource distribution between urban and rural areas. Attention should be paid to the temporal and spatial characteristics of urban and rural water resources, but this problem has not been paid enough attention. Most studies on the dynamic relationship between urban and rural water resources have focused on the qualitative research level, and no studies have directly measured the coupling coordination level of urban and rural water resources, the historical context or the lack of attention that has been paid to regional differences. The Chinese government has pointed out that “on the basis of continuing to promote development, efforts should be made to solve the problem of unbalanced and inadequate development”. Therefore, it is necessary to further grasp the regional differences and sources of the coupling and coordination of water resources between urban and rural areas in China, comprehensively analyze the current situation and underlying causes of regional unbalanced development, provide a factual basis for promoting regional coupling and coordination development and provide reference ideas for solving the problems of unbalanced and inadequate development. Finally, existing studies lack a reasonable explanation of the driving factors of the interaction between urban and rural water resources. Although China has introduced relevant water resource management policies and obtained successful cases, as presented by the authors in [22], there is still a lack of practical experience in improving urban and rural water resource systems in different regions, resulting in a persistent contradiction between urban and rural water resource allocation. Studies on the influencing factors of urban and rural water resources are limited to a certain degree in urban or rural areas, and they lack investigations into the overall impact of urban and rural water resources. Therefore, an in-depth discussion on the mechanism of influencing factors is needed to further investigate the possible regional heterogeneity of influencing factors. The contribution of this manuscript is mainly reflected in three aspects: First, it discussed the relationship between urban and rural water poverty. Existing studies have mainly focused on basin or rural water poverty, ignoring the interaction between urban and rural water resources. Therefore, from the perspective of water poverty, this paper used the coupled coordinated development model to determine the relationship between urban and rural water poverty, based on the inheritance and expansion of the existing research. Second, for the study of urban and rural water poverty, this paper further considered the influence of spatial interaction, that is, exploring the interaction between regions, and expanding upon the application of the spatial econometric model in this field. Third, although there have been studies on the influencing factors of rural water poverty in the academic community, they have mainly focused on the consideration of dimensions or indicators. This paper introduced the Tobit model to consider the influencing factors of the interaction relationship between urban and rural water poverty through a regression analysis, which constitute the blind spot of empirical analysis in this field. We believe that our study makes a significant contribution to the literature because we analyzed urban and rural water shortages within the same framework, which is lacking in the current literature. Based on the above background, according to the relative weakness of current domestic water poverty research, this paper started by measuring the actual situation of water resources in northwest China, designed an evaluation framework suitable for urban and rural water resources in this region and measured urban and rural water poverty in 52 regions in northwest China from 2000 to 2020. On this basis, we scientifically described the space–time evolution characteristics of the coupled and coordinated development of urban and rural water poverty, and we analyzed the internal influence mechanism of the coupled and coordinated development of urban and rural water poverty, in order to provide a theoretical basis and policy suggestions for alleviating the current urban–rural water contradiction in the “urban–rural dual structure” model in northwest China.

## 2. Methodology and Study Area

### 2.1. Urban and Rural Water Poverty Index

Based on the theory of water poverty, this paper adopted the comprehensive weighting method to construct the WPI model. The WPI model, namely, the water poverty index, evaluates the degree of water shortage from five dimensions: resource (R), facilities (F), capacity (C), use (U) and the environment (E). Among them, resources refer to the quantity of water resources and their availability in the physical sense; facilities refer to the conditions of sufficient and complete water supply, irrigation and sanitation facilities, which reflect the utilization degree of clean water resources and the safety degree of water resource utilization of residents in the region; capacity refers to the ability to evaluate residents’ comprehensive utilization and management of water resources and sanitation facilities on the basis of economic, educational and health conditions; use refers to measuring the contribution of water resources to social and economic development by analyzing the water consumption of various sectors of society; and the environment refers to the environmental status related to water resource management, including water quality status and potential pressure on the ecological environment [23]. The calculation formula is as follows:(1)$$WPI=w_{r}\times resource+w_{a}\times access+w_{c}\times capacity+w_{u}\times use+w_{e}\times environment$$
where $w_{i}$ refers to weight; resource, access, capacity, use and environment represent the weighted scores of the indicators of five dimensions after data standardization; WPI is the score of water poverty. The lower the value, the more serious the shortage of water resources [19,21].

If the WPI index is used to evaluate the shortage of urban and rural water resources, the actual situation in northwest China must be considered more comprehensively. In terms of previous research results, the selection of the WPI index system has been relatively rich and perfect. However, considering the research perspective of this paper, on the premise that the total amount of water resources is established, the selection of urban and rural water poverty evaluation indicators in this paper not only referred to the WPI index system, but also integrated the advantages of WPI index selection [24,25], and focused on the allocation right and use ability of water resources. We mainly followed the scientific principles of particularity, accessibility and a strong contrast, deleting some indicators that have no comparative value, and enriching some indicators that are able to compare and reflect the water use situation of urban and rural areas, so that the existing WPI index system is more in line with the actual situation of urban and rural water resources in northwest China.

First of all, the allocation of urban and rural water resource rights was considered in terms of resources. Secondly, water facilities not only included the facilities that reflect the water supply capacity of production and domestic water, but also considered the facility level of drainage capacity; thirdly, government support, residents’ water use and water intake capacity and residents’ awareness of water conservation can reflect the utilization capacity and thus they were adopted to enrich the original index system. Finally, we comprehensively evaluated the use of water resources, and refined this into indicators of domestic water, production water and environmental protection. Whether the weighting method is reasonable or not directly affects the accuracy of the final result. Therefore, this paper adopted a combination of subjective and objective methods to weigh the indicators, in which the subjective weight adopted an analytic hierarchy process (AHP), the objective weight adopted a principal component analysis (PCA) and the comprehensive weight took the average value of the two. The results are shown in Table 1.

### 2.2. Coupling Coordination Degree Model

Referring to the capacity coupling coefficient model in physics, the coupling degree model of urban and rural water poverty in northwest China was constructed as follows:(2)$$C={\left\{\frac{U_{1}\times U_{2}}{{\left(U_{1}+U_{2}\right)}^{2}}\right\}}^{1/2}$$
where C is the coupling degree, and the value range is [0, 1]. The larger the value, the higher the correlation degree between systems; $U_{1}$ and $U_{2}$ represent the scores of urban water poverty and rural water poverty, respectively. The degree of coupling can only indicate the strength of the interaction between urban water resources and rural water resource systems and cannot measure the degree of coordination of development between systems. Therefore, the coordination degree model was introduced to measure the coordinated development of urban and rural water resource systems, and the calculation formula was as follows:(3)$$D={\left(C\times T\right)}^{1/2},\;T=\left(\alpha \times U_{1}+\beta \times U_{2}\right)$$
where D is the coordination degree, and T is the comprehensive coordination index of urban and rural water poverty. Considering that the two contribute equally to the coordinated development of urban and rural water poverty in the process of water resource development, both α and β values were 0.5. According to the coordination degree D and the level of urban water poverty $U_{1}$ and rural water poverty $U_{2}$, and drawing on the division of coordination types in physics, the coupling coordination types of urban and rural water poverty were divided into 3 categories, 4 subcategories and 12 coupling coordination types (Table 2).

### 2.3. Spatial Autocorrelation Model

Spatial autocorrelation is a common method used to analyze the spatial heterogeneity and correlation of data, which are used to explain the correlation characteristics of the spatial attribute data. The global Moran’s I index was used to measure the spatial correlation characteristics of the urban–rural water poverty coupling coordination degree in northwest China, and its spatial agglomeration situation was analyzed [20]. The calculation formula was:(4)$$Global\;Moran^{\prime }s\;I=\frac{\sum_{i}^{n}\sum_{j\neq 1}^{n}w_{ij}\left(x_{i}-\bar{x}\right)\left(x_{j}-\bar{x}\right)}{S^{2}\sum_{i}^{n}\sum_{j\neq 1}^{n}w_{ij}}\;$$

Moran’s I range is [−1, 1], and when it is a positive number, it indicates that the urban–rural water poverty coupling coordination degree is positively correlated in space (the elements with the same type of attribute value are adjacent and close). When it is 0, it means that the spatial distribution of the urban–rural water poverty coordination degree is random, and there is no spatial correlation. When it is negative, it indicates the spatial negative correlation of the urban–rural water poverty coupling coordination degree (the elements of different types of attribute values are adjacent and close).

Local spatial autocorrelation was used to identify the spatial correlation mode formed by the urban–rural water poverty coupling coordination degree between regions, so as to find the spatial heterogeneity characteristics of urban–rural water poverty coupling coordination development in northwest China. The calculation formula was:(5)$$I_{i}=\left(x_{i}-\bar{x}\right)\times \overset{n}{\underset{j\neq 1}{\sum }}w_{ij}\left(x_{j}-\bar{x}\right)$$

When $I_{i}>0$, it means that the regions with the same type of element values are close to each other, which shows a high–high aggregation (H–H) or low–low aggregation (L–L) of the urban–rural water poverty coupling coordination degree; when $I_{i}<0$, it means that the regions with different types of element values are close to each other, that is, there are spatial outliers, which are manifested as a high–low aggregation (H–L) and low–high aggregation (L–H) of the urban–rural water poverty coupling coordination degree.

### 2.4. Hot Spot Analysis

The hot spot analysis is a method to determine the distribution characteristics of local spatial clustering which is used to measure the clustering relationship between each unit and its surrounding units. Getis-Ord $G_{i}^{*}$ was used to measure the spatial distribution of hot spots and cold spots of the urban and rural water poverty coupling coordination degree. The calculation formula was:(6)$$G_{i}^{*}\left(d\right)=\overset{n}{\underset{i=1}{\sum }}w_{ij}\left(d\right)\mu_{i}/\overset{n}{\underset{i=1}{\sum }}\mu_{i}$$
(7)$$Z=\frac{G_{i}^{*}-E\left(G_{i}^{*}\right)}{\sqrt{Var\left(G_{i}^{*}\right)}}$$
where $G_{i}^{*}$ is the agglomeration index of spatial unit *i*, and Z is the significance of the agglomeration index. The higher the Z score, the closer the high-value spatial agglomeration (hot spots) of the urban–rural water poverty coupling coordination degree; on the contrary, it shows that the low-value space agglomeration (cold spots) of the urban–rural water poverty coupling coordination degree is closer.

### 2.5. Selection of Variables

The coupling development of urban and rural water poverty is affected by many factors. Referring to the existing research [27,28,29,30,31,32,33,34,35] and combining this with the actual situation, the index system was constructed from the degree of industrialization, the level of opening-up, technological progress, population size, the level of economic development, industrial structure, labor transfer, agricultural expenditure, agricultural production level, environmental regulation, etc. (Table 3). The coupling coordination degree of urban and rural water poverty changed between 0 and 1, and the explained variables were truncated, which conformed to the setting conditions of the Tobit regression model of the restricted dependent variable. In this paper, the random effect panel Tobit model was used for the econometric estimation. On the one hand, compared with the fixed effect panel Tobit model, the random effect panel Tobit model can obtain a consistent estimation [36]. On the other hand, it can effectively avoid biased results caused by least square regression [37]. The model was set as follows:(8)$$D_{it}=cons+\beta_{1}indu_{it}+\beta_{2}open_{it}+\beta_{3}tech_{it}+\beta_{4}per_{it}+\beta_{5}pgdp_{it}+\beta_{6}is_{it}\;+\beta_{7}labtra_{it}+\beta_{8}fina_{it}+\beta_{9}pgrain_{it}+\beta_{10}er_{it}+\varepsilon_{it}$$
where $D_{it}$ represents the degree of coupling coordination, i represents region, t represents time; cons represents constant term; indu refers to the degree of industrialization; open represents the level of opening-up; tech represents technological progress; per represents population size; pgdp represents the level of economic development; is represents industrial structure; labtra represents labor transfer; fina represents agricultural expenditure; pgrain represents the level of agricultural production; er represents environmental regulation; and $\varepsilon_{it}$ represents the random disturbance term. Using Stata 15.1 econometric analysis software (TurnTech LLC, Beijing, China), the random effect panel Tobit regression was carried out, and the results are shown in Section 3.6.

### 2.6. Study Area

Northwest China belongs to the dry early morning and semi-dry early morning region of China, which mainly includes Xinjiang, Ningxia, Qinghai, Gansu, Shaanxi and other provinces (Figure 1). The land area accounts for one-third of the total area of China, and the population accounts for approximately 7% of the total population of the country. However, the water resources only account for approximately 4% of the total water resources of the country, and the per capita water resources only account for 80% of the per capita level of the country. With the growth in population and economic development, the water consumption and the demand for water increase, and the exploitation and utilization of water resources gradually increase. At the same time, there is a serious waste of water resources. Due to natural and human-made reasons, the contradiction between the supply and demand of water resources is more prominent, resulting in a serious shortage of ecological, industrial, agricultural and urban water, resulting in dry water surfaces, shrinkage of the oasis, riverbed flow interruption, water pollution and deterioration of the ecological environment in northwest China. At present, the ecological environment in northwest China is extremely fragile due to drought and water shortages. However, the people’s exploitation of water resources for production, living and survival has not decreased, which has greatly exceeded the carrying capacity of water resources in the region. In addition, people’s awareness of ecological environment protection is not strong. As a result, the region is experiencing a water resource shortage, production development is blocked and water for economic development is crowding out the ecological water situation. The water resource shortage has become the restrictive factor of the development and sustainable development of northwest China. 

Firstly, northwest China is located inland, far away from the ocean and cut off by mountains. This region experiences little precipitation, and the limited precipitation is mainly concentrated in the summer and autumn; the summer water shortage is very serious. Not only do water resources fail to meet the needs of agricultural irrigation and industrial production during a shortage, but people in many places even have trouble reserving water. In northwest China, there are only 30 large reservoirs, which have a low regulation of water volume and engineering water shortage, resulting in severe agricultural spring drought and a greatly reduced grain output; this even means that grain self-sufficiency is not possible. For example, at present, diversion projects with a low control capacity are still the main means of water supply in Xinjiang, and the water supply capacity of regulatory reservoir projects only accounts for 14.4% of the total water supply capacity; the plain reservoirs are the main reservoirs, while the controlled reservoirs in mountainous areas are very low in scale, resulting in a situation of low drought resistance ability and poor spatial and temporal regulation ability. The spring irrigation period results in a serious water shortage, and the flood season irrigation not only results in wastewater, but also aggravates the secondary salinization of land. As another example, in the existing water supply projects in Shaanxi, there are few water storage projects. In large rivers in the Guanzhong area, where water shortage is serious, three-quarters of farmland irrigation can be supplied by diversion projects, and the degree of water supply assurance is very low.

Secondly, the spatial and temporal distribution of water resources in northwest China is more uneven, which means it is difficult to use effectively. At the same time, as the temperature rises, glaciers and mountain snow melt away, lakes and wetlands shrink, evaporation increases and available water sources in different regions basically decrease further. The surface runoff is mainly concentrated in the flood season, accounting for more than 60% of the annual runoff. It is precisely because of the influence of this precipitation characteristic that the utilization of water resources is greatly restricted and disasters are frequent, resulting in the instability of industrial and agricultural production. The northwest region has a lack of water resources, and most of the regions west of Lanzhou, namely, Ri, Ning, Inner Mongolia, Qing and Xin, have an annual precipitation of less than 300 mm. For more than 1 million square kilometers of desert and the Gobi area, annual rainfall is as low as 50 mm or less. The air is dry and the evaporation is vigorous. Drought is defined when the dryness index (the ratio of evaporation capacity to precipitation) is greater than 1. In northwest China, it can be greater than 50. Water and life have the same significance. In the past 20 years, the annual rainfall has increased in some parts of northwest China, but the spatio-temporal distribution has been more uniform. 

Thirdly, the water efficiency of industry and agriculture in northwest China is low. Industrial water technology is backward and the reuse rate of water is very low. For example, the total water consumption per CNY ten thousand of output value of the main industries in Lanzhou, Gansu is 370 t; Urumqi, Xinjiang is 707 t; and Xining, Qinghai is 1764 t. These are 3.8 times, 7.3 times and 18 times that of water-saving Qingdao, Shandong (97 t), respectively. Agricultural water waste is even more shocking, accounting for more than 70% to 90% of the total water consumption. Some old irrigation areas have, for a long time, used flood irrigation and string irrigation; however, field projects do not match and canal system leakage is serious; the effective benefit coefficient of the canal system is generally 0.4, and the highest is 0.54. In terms of extensive agricultural management, the irrigation quota is too high.

### 2.7. Data Sources

The study took 52 regions in northwest China as the research object, and the index data used were mainly from the 2000–2020 “China Environmental Statistical Yearbook”, “China Urban Construction Statistical Yearbook”, the statistical yearbook of five provinces in northwest China and related cities and the statistical bulletin of water resources. Some missing data were obtained by using the average value of adjacent years or the fitting prediction value of different periods.

## 3. Result and Analysis

### 3.1. Measurement of Urban Water Poverty

Formula (1) was used to calculate the urban water poverty score and spatial-temporal distribution pattern in northwest China, as shown in Figure 2. In 2000 and 2020, the average score of urban water poverty was 0.2390 and 0.2558, the standard deviation was 0.0366 and 0.0345 and the coefficient of variation was 0.1530 and 0.1347, respectively. While the degree of urban water poverty showed a downward trend, the absolute gap and relative gap between regions showed a narrowing trend, indicating that the overall development of regional urban water resources was going in a balanced direction. However, the improvement speed of water resources in regions in northwest China was not the same, and an obvious gap existed. According to the “natural breaks” method (Jenks), the urban water poverty score is divided into four levels: very low-level areas, low-level areas, medium-level areas and high-level areas. In 2000, urban water poverty in 21 regions, including Ankang, Lanzhou, Kezilesu, etc., was at a medium and high level, accounting for 40.4% of all regions, mainly distributed in the northern and southern regions of the northwest region. Urban water poverty in 31 regions, including Haixi Prefecture, Bayinguole, Jiuquan, etc., was at a very low and low level, accounting for 59.6% of all regions, widely distributed in the central and eastern regions of the northwest region. In 2010, urban water poverty in 25 regions, such as Xi’an, Lanzhou, Yinchuan, etc., was at a medium and high level, accounting for 48.1% of all cities, and showing great changes in quantity and spatial distribution, and high-value ethnic groups formed near Guoluo, Xi’an and other regions. Urban water poverty in 27 regions, including Linxia, Hainan, Dingxi, etc., was at a very low and low level, accounting for 51.9% of all regions, concentrated in the central and northern regions of the northwest region. In 2020, urban water poverty in six regions, including Xi’an, Lanzhou and Urumqi, etc., was at a medium and high level, mostly provincial capital cities, accounting for 11.5% of all cities. Urban water poverty in 46 regions, including Kezilesu, Aletai, and Hetian, was at a very low and low level, accounting for 88.5% of all regions, and widely distributed in the western and northern regions of the northwest region. From 2000 to 2020, areas with high urban water poverty scores gradually moved eastward, while areas with low urban water poverty scores gradually moved westward, and the “east–central–west” stepped pattern was obvious.

### 3.2. Measurement of Rural Water Poverty

Formula (1) was used to calculate the score of rural water poverty and the spatial-temporal distribution pattern in northwest China, as shown in Figure 3. In 2000 and 2020, the average score of rural water poverty was 0.2002 and 0.2419, the standard deviation was 0.0348 and 0.0518 and the coefficient of variation was 0.1737 and 0.2140, respectively. While the development of rural water resources had significantly improved, the absolute gap and relative gap between regions showed an expanding trend, indicating that the rural water resource system had further improved. However, there remained a great difference in the improvement speed of rural water resources in northwest China. The two-level differentiation was gradually intensifying, and the difference between the maximum value and the minimum value was nearly three-fold. According to the “natural breaks” method, the rural water poverty score was divided into four levels: very low-level areas, low-level areas, medium-level areas and high-level areas. In 2000, the rural water poverty in 15 regions, including Xi’an, Baoji, Yili, etc., was at a medium and high level, accounting for 28.8% of all regions, and mainly distributed in the eastern and northern regions of northwest China. Rural water poverty in 37 regions, including Jiayuguan, Jinchang Zhangye, etc., was at a very low and low level, accounting for 71.2% of all cities, and widely distributed in the central region of northwest China. In 2010, rural water poverty in 23 regions, including Xianyang, Weinan and Yanan, was at a medium and high level, accounting for 44.2% of all regions, with an increase in quantity, but little change in spatial distribution. Rural water poverty in 29 regions, including Baiyin, Tianshui, Wuzhong, etc., was at a very low and low level, accounting for 55.8% of all cities. The number has decreased, but the spatial distribution has not changed much. In 2020, rural water poverty in 21 regions, including Lanzhou, Yinchuan, Xining, etc., was at a medium and high level, accounting for 40.4% of all cities, and high-value ethnic groups formed near the provincial capital cities. Rural water poverty in 31 regions, including Gannan, Yushu, Bayinguole, etc., was at a very low and low level, accounting for 59.6% of all regions, and widely distributed in the central region of northwest China. From 2000 to 2020, areas with high rural water poverty scores gradually moved southeast, and areas with low rural water poverty scores gradually moved northwest, showing a collapse pattern of “high on both sides and low in the middle”.

### 3.3. The Measurement of the Coupling Coordinated Development Degree

In order to explore the interactive relationship between urban and rural water resource systems, the urban–rural water poverty coupling coordination in northwest China was calculated using Formula (3) and combined with Table 3. In addition, the spatio-temporal distribution pattern of urban–rural water poverty coupling coordinated development is shown in Figure 4. In 2000, the coupling coordination level of urban and rural water poverty was low on the whole. Only Xi’an, Ankang and Turpan belonged to the basic coordination type, accounting for 5.8% of all cities. Urban and rural water poverty in 49 regions, including Tongchuan, Baoji, Xianyang, etc., was basically uncoordinated, accounting for 94.2% of all regions. At the same time, there remained heterogeneity in the classification of the subclasses among regions. Among them, Lanzhou, Jiayuguan, Guoluo and four other regions belonged to the “basically uncoordinated, rural lagging type”. Yanan belonged to the “basically uncoordinated, urban lagging type”. In 2010, the coupling coordination level of urban and rural water poverty had improved on the whole. The number of basic coordination areas had increased to six, accounting for 11.5% of all regions. The areas that were added included Weinan, Yanan, Yulin and Yili, and the area that was no longer included was Turpan. The number of basically uncoordinated areas decreased to 46, accounting for 88.5% of all regions. Among them, Guoluo and Yushu were still the “basically uncoordinated, rural lagging type”, indicating that the speed and degree of improvement in urban water resource systems were much faster than those of rural areas, and the gap between urban and rural water conservancy facility construction was widening. In 2020, the coupling coordination level of urban and rural water poverty had greatly improved, and the number of basic coordination areas had increased to 15, accounting for 28.8% of all cities. The areas that were added were Baoji, Xianyang, Hanzhong, Lanzhou, Tianshui, Pingliang, Guyuan, Xining and Changji, while the area that was no longer included was Kashgar.. Among them, Hanzhong belonged to the “basically coordinated, urban lagging type”, which shows that the improvement speed and degree of the rural water resource system were able to catch up with that of city areas, and the rural water pollution control ability had greatly improved. The number of areas with basic uncoordinated types decreased to 37, accounting for 71.2% of all regions. From 2000 to 2020, the coupling coordinated development level of urban–rural water poverty in northwest China has experienced a slow rise from 2000 to 2010 and a rapid rise from 2010 to 2020. However, nearly 70% of the regions were still basically uncoordinated. The overall situation of the coupling coordinated development relationship between urban water poverty and rural water poverty in northwest China is not optimistic.

### 3.4. Spatial Autocorrelation of the Coupling Coordinated Development Degree

In order to further study the spatial differentiation characteristics of the coupling coordinated development of urban and rural water poverty, the global Moran’s I indices of the coupling coordinated degree of urban and rural water poverty in 2000, 2010 and 2020 were calculated by ArcGIS to be 0.122, 0.288 and 0.251, respectively. All were positive and passed the significance test with a confidence of 5%, indicating that there was an obvious spatial agglomeration phenomenon between similar observations of the coupling coordinated degree of urban and rural water poverty in northwest China. The global Moran’s I index showed a trend of first increasing and then decreasing, indicating that the spatial agglomeration of the coupling coordinated development of urban and rural water poverty in northwest China first increased and then decreased, and the heterogeneity first decreased and then increased. In the case of regional differences in terms of the overall spatial heterogeneity, local spatial differences may expand [26]. In order to comprehensively reflect the change trend in regional differences and explore whether there was an autocorrelation relationship between local research units in the adjacent space in the coupling coordination degree of urban and rural water poverty, a Moran scatter diagram was used to analyze the local spatial correlation and spatial difference degree of urban and rural water poverty coupling coordination of prefecture level units in different years, and the agglomeration and distribution of Lisa were obtained, as shown in Table 3. The types of spatial agglomeration can be divided into the following categories:

① High–high (H–H): The coupling coordination degree between itself and the surrounding areas was high, and the spatial difference was small. In 2000, this type of region included 12 regions such as Xi’an, Turpan, etc. In 2010, it included 10 regions such as Shangluo, Akesu, etc. By 2020, it had evolved into 11 regions such as Tianshui, Pingliang, etc. The number change was relatively stable, accounting for approximately 21.2% of the total in northwest China. These areas belonged to the growth pole of water use, and gradually moved to southern Shaanxi and central Gansu during the study period. Specifically, in 2000, this type of area was concentrated in northern Xinjiang and central and southern Shaanxi, forming a high-value agglomeration area centered on Xi’an and Turpan. In 2010, the number of high–high clusters in northern Xinjiang decreased to one, and the number of high–high clusters in southern Shaanxi increased to nine. In 2020, the southeast of northwest China still formed high–high agglomeration areas, and south Shaanxi was the region with the highest number of high–high agglomeration types, which was also the most stable during the study period. Tianshui and Pingliang near the central part of Shaanxi Province also formed a significant high–high concentration, which showed that the regional growth pole centered on Xi’an had a significant spillover effect in factor flow, reward transfer and technology diffusion, thus driving the continuous coupling coordinated development of urban and rural water poverty in the surrounding areas.

② Low–low (L–L): The coupling coordination degree between itself and the surrounding areas was low, and the spatial difference was small. In 2000, this type of region included nine regions such as Qingyang, Wuwei, etc. In 2010, it included 10 regions such as Wuzhong, Hotan, etc. By 2020, it had evolved into 12 regions such as Longnan, Dingxi etc., showing a slow growth trend, and accounting for about 19.9% of the total in the northwest region, mainly stably concentrated in eastern Gansu, central Xinjiang and northern Ningxia. This was basically consistent with the spatial distribution of contiguous poverty-stricken areas such as the Qinba mountain area and the Liupan mountain area. The development of urban and rural water resources was greatly constrained by restrictive conditions. These were the “water shortage” agglomeration areas with low scores of urban and rural water poverty and a slow growth rate, which formed an obvious negative conduction effect in space, seriously restricting the development process of urban and rural water resources, so the coupling coordination degree was low. Although there is a lot of room for improvement, it is difficult to improve the coupling coordination degree of urban and rural water poverty simply by relying on its own development in a short period of time.

③ High–low (H–L): The self-coupling coordination degree was high, but the coupling coordination degree of the surrounding areas was low, and the spatial difference was large. In 2000, this type of region included nine regions such as Lanzhou, Xining, Urumqi, etc. In 2010, it included 10 regions such as Yinchuan, Guyuan, Yili Hasake Autonomous Prefecture, etc. By 2020, it evolved into nine regions such as Zhangye, Jiuquan, Changji Hui Autonomous Prefecture, etc. The number has changed steadily, accounting for approximately 17.3% of the total in the northwest region. This type of area covers some provincial capitals, and gradually moved to northern Xinjiang and northern Gansu during the study period. Although the coupling coordinated development degree of urban–rural water poverty in such areas was at a high level, there was still a certain difference compared with the growth polar region, and there was still a large growth space for the coupling coordinated development degree of urban–rural water poverty. However, the improvement in urban–rural water poverty coupling coordination in these areas did not drive the improvement in the coupling coordination level in the surrounding areas, but showed a polarization phenomenon, which restricted the coupling coordinated development of urban–rural water poverty to a certain extent.

④ Low–high (L–H): The self-coupling coordination degree was low, but the coupling coordination degree of the surrounding areas was high, and the spatial difference was large. In 2000, this type of region included 20 regions such as Jiayuguan, Jinchang, Linxia Prefecture, etc. In 2010, it included 20 regions such as Baiyin, Wuwei, Hami, etc. By 2020, it had evolved into 18 regions such as Haidong, Yushu Prefecture, Shizuishan, etc. The quantitative change and spatial distribution were relatively stable, accounting for approximately 37.2% of the total in the northwest region. It belonged to the “transition area”, from the area with a good coupling coordination degree of urban and rural water poverty to the area with a poor coupling coordination degree of urban and rural water poverty. Specifically, these regions had limited natural resources and economic and social capabilities, and their urban–rural water poverty coupling coordination degree was low, with slow growth. Most of them were distributed around high–low agglomeration areas. The positive and negative interaction between the two sides weakened the regional spillover transmission effect to a certain extent, weakening the urban–rural water poverty coupling coordination development in this region.

### 3.5. Hot Spot Analysis of the Coupling Coordinated Development Degree 

In order to explore the path to optimize the allocation of urban and rural water resource elements in northwest China, the Getis-Ord $G_{i}^{*}$ of the urban and rural water poverty coupling coordination degree in 2000, 2010 and 2020 was analyzed by using ArcGIS. According to the “natural breaks” method (Jenks), the scores of Getis-Ord $G_{i}^{*}$ were layered from high to low, and the distribution map of cold and hot spots of the urban and rural water poverty coupling coordination degree was generated (Table 4). Through a comparative analysis, the following can be concluded: ① In 2000, the hot spots of the urban–rural water poverty coupling coordination degree in northwest China contained 10 regions, including Xi’an, Tongchuan and Baoji; the cold spots contained 8 regions, including Lanzhou, Baiyin and Wuwei. ② In 2010, there was no change in the hot spots of the urban–rural water poverty coupling coordination degree in northwest China. The number of cold spots increased to 14, and the areas that were added were Huangnan, Gannan, Xining, Jinchang, Hami and Bayinguole. The cold spots formed a spatial agglomeration trend in the middle of the northwest region. ③ In 2020, the number of hot spots increased to 14, and the areas that were added were Tianshui, Pingliang, Longnan and Guyuan. The number of cold spots decreased to eight, the areas that were no longer included were Lanzhou, Baiyin, Wuwei, Jiuquan, Linxia, Gannan, Haidong, Huangnan and Hami and the areas that were added were Jiayuguan, Kezilesu and Kashgar. Overall, from 2000 to 2020, the spatial cold and hot spots of the coupling coordinated development of urban and rural water poverty in northwest China changed significantly, showing a spatial pattern evolution trend in the transition from a low-significance area to a medium-significance area of the hot spots, and an upgrade from the medium-significance area to a high–significance area of the hot spots. The hot spots in the southeast were gradually expanding. It can be inferred that as long as the city is in a hot spot, the coupling coordination degree of urban and rural water poverty in the surrounding areas will also be high. 

### 3.6. Analysis on Driving Factors

#### 3.6.1. Total Regression Analysis

As shown in the Table 5, there was a positive correlation between technological progress and the degree of coupling coordination, which was significant at the 1% level, indicating that increasing research and experimental development (R&D) expenditure can improve the efficiency of water resource utilization by improving the total factor productivity of enterprises and the treatment rate of wastewater and sewage, and further help to promote the coupling coordinated development of urban and rural water poverty. There was a positive correlation between the level of economic development and the degree of coupling coordination, which was significant at the 1% level. Regional economic growth, on the one hand, means more consumption of water resources, which may have a certain negative impact on water use efficiency. On the other hand, it also means that the innovation speed of water-saving technology is accelerated, so the efficiency of production and domestic water use is higher. The regression coefficient was significantly positive, indicating that economic development improves the efficiency of water resource utilization and the ability of water pollution prevention and control, and makes up for the negative effects of water consumption. The regression coefficient of the industrial structure was positive and significant at the level of 1%, indicating that the increase in the proportion of the tertiary industry has a positive impact on the coupling coordinated development of urban and rural water poverty. Different industrial structures determine the efficiency of urban and rural water resource utilization to a certain extent. Therefore, optimizing the industrial structure and increasing the proportion of the tertiary industry are important driving forces for the coupling coordinated development of urban and rural water poverty in northwest China. The regression coefficient of agricultural expenditure was positive and significant at the 1% level, indicating that in the process of realizing the coupling coordinated development of urban and rural water poverty, we should not only rely on the decisive role of the “invisible hand” of the market in resource allocation, but also effectively play the macro-control role of the “visible hand” of the government. The increase in financial support for agriculture, forestry and water is conducive to improving the infrastructure of farmland and water conservancy, improving the agricultural production environment and improving the efficiency of agricultural water use, which has become an important part of promoting the coupling coordinated development of urban and rural water poverty in northwest China. The regression coefficient of the agricultural production level was positive and significant at the 1% level, indicating that a certain scale of cultivated land for grain crop planting will increase the production factors per unit area, which is conducive to the large-scale production of grain. Changing the extensive management mode to the intensive mode will not only help to consolidate soil and water conservation, and promote the efficiency of soil and water loss control, but also help to accelerate the coupling coordinated development of urban and rural water poverty in northwest China.

#### 3.6.2. Regional Regression Analysis

Further, according to the administrative division, this paper divided the northwest region into five provinces, Shaanxi, Gansu, Ningxia, Qinghai and Xinjiang, in order to identify the driving factors of different regions in more detail, and then formulate targeted urban and rural water resource management policies. It can be seen from Table 5 that the regression coefficient of the industrialization degree of Ningxia and Xinjiang was negative, and passed the significance test of 1% and 5%, indicating that the industrialization process has a negative impact on the coupling coordinated development of urban and rural water poverty. The industrial characteristics of high pollution, high energy consumption and high emission in the two regions are obvious, so it is urgent to speed up the adjustment of the industrial structure and change the development mode. The regression coefficient of Shaanxi and Xinjiang’s opening-up level was positive, while the regression coefficient of Gansu and Ningxia was negative, and both passed the significance test of 5% and 10%, indicating that Shaanxi and Xinjiang urgently need to expand their opening-up; attract a large number of foreign capital, technology and talents; drive the optimization of industrial structure; improve water efficiency; and promote the coupling coordinated development of urban and rural water poverty. Gansu and Ningxia should control foreign investment and avoid focusing on low-value-added and high-polluting industries, so as to promote the coupling coordinated development of urban and rural water poverty. The regression coefficient of technological progress in Gansu and Xinjiang was positive, and passed the significance test of 1% and 5%, indicating that it is necessary to further strengthen the investment in research and experimental development, speed up the innovation of water-saving technology and gradually improve the utilization efficiency of water resources. The regression coefficient of the population size of Shaanxi and Xinjiang was positive, and the regression coefficient of Qinghai was negative, and both passed the significance test. According to the classification of the water resource carrying capacity proposed by scholars [38], in 2020, Shaanxi belonged to the warning category of the water resource population carrying capacity, and Xinjiang and Qinghai belonged to the surplus category of the water resource population carrying capacity, indicating that the water resources of Shaanxi and Xinjiang have the ability to ensure a large degree of population growth, and Qinghai has a surplus of water resource population carrying capacity. However, due to the low efficiency of water resource utilization, this will not lead to the coupling coordinated development of urban and rural water poverty. The regression coefficient of the economic development level and industrial structure has always been positive, and both passed the significance test, indicating that the improvements of regional economic strength, reasonable industrial structure, perfect infrastructure and advanced technology have effectively promoted the coupling coordinated development of urban and rural water poverty. The regression coefficient of the labor transfer in Qinghai was negative, which was significant at the 1% level, and the impact of the other four provinces was not significant. This indicated that, after the transfer of rural labor to cities in Qinghai, the hollowing-out and aging of rural areas had been aggravated, and the rural population left behind failed to effectively use the abundant production and resource space, or reduce their own water rights and water capacity, which has hindered the coupling coordinated development of urban and rural water poverty. The regression coefficients of agricultural expenditure in Gansu, Ningxia, Qinghai and Xinjiang were positive, and all passed the significance test at the 1% level, while the regression coefficient in Shaanxi was negative, and significant at the 5% level, indicating that in the process of implementing the combined fiscal policy of financial support for agriculture (including productive and non-productive expenditure), on the one hand, we should strengthen the input of productive expenditure and narrow the gap between urban and rural water resource allocation. On the other hand, we should strengthen the management of non-productive expenditure and prevent “capital rent-seeking” behavior, which will eventually sacrifice the interests of farmers and lead to policy distortion. The impact of the agricultural production level on Shaanxi was not significant, while the regression coefficient of the other four provinces was positive, and all of them passed the significance test, indicating that the land suitable for agricultural development should be intensively managed to provide a good external environment for the efficient utilization of urban and rural water resources. The regression coefficient of environmental regulation in Shaanxi was positive and that in Xinjiang was negative, and both were significant at the 1% level, indicating that there is regional heterogeneity in the impact of environmental regulation. On the one hand, this will lead to the reduction in regional wastewater discharge, which is conducive to the coupling coordinated development of urban and rural water poverty; on the other hand, it may increase the production cost of enterprises and have a certain negative impact on the economy, which is not conducive to the coupling coordinated development of urban and rural water poverty.

## 4. Discussion

Based on the above research conclusions, this paper proposes the following urban and rural water resource management policies: On the one hand, the top-level design of northwest China needs to be strengthened in the future, by accelerating the transformation and upgrading the industrial structure, changing the development mode of high energy consumption and high emission industries and further enhancing the regional economic strength. For regions with a high grain yield per unit area, the agricultural planting and management mode needs to be improved, promoting large-scale and scientific production methods, and improving the efficiency of water resource utilization. At the same time, the financial expenditure on supporting agriculture should be optimized, establishing a multi-channel financing mode of water conservancy facilities led by the government to be participated in by all sectors of society, and ensuring the continuity and stability of investment in water conservancy infrastructure. The sewage treatment system should be improved; the monitoring of the water supply quality and environmental pollution should be strengthened, such as increasing the number of monitoring points and water quality monitoring indicators of important rivers; and the monitoring frequency and information transparency should be increased, so as to ensure a timely and comprehensive grasp of the water quality status of major river basins. The awareness of water resource protection should be improved, and people should be made deeply aware of the importance of water resource protection by popularizing the laws and regulations related to water resources and the consequences of water pollution. On the other hand, regional water resource management policies should be implemented according to local conditions. Shaanxi should continue to maintain the trend in rapid growth in the coupling coordinated development of urban and rural water poverty, giving play to the advantages of spatial agglomeration, improving the radiation driving effect, improving the regional environmental monitoring network, grasping the strength and nodes of environmental regulation and avoiding the negative impact of the “one size fits all” model on the coupling coordinated development of urban and rural water poverty. Gansu and Ningxia should appropriately raise the environmental access threshold for the introduction of foreign capital, strictly review the green production standards of foreign capital, pay attention to cultivating local alternative industries, promote the integrated governance standards of domestic and foreign capital, require enterprises with serious pollution emissions to formulate green transformation plans and impose high environmental taxes on those that are overdue. Qinghai should further optimize the industrial structure, increase financial support for the construction of farmland and water conservancy facilities, improve the efficiency of urban and rural water resource utilization, formulate reasonable urban population flow policies, promote low-water consumption household equipment and improve residents’ awareness of water resource crises through continuous and extensive publicity, so as to promote the coupling coordinated development of urban and rural water poverty. Xinjiang should increase the R&D expenditure on water-saving irrigation, wastewater treatment and other technologies and equipment, and give play to the spillover effect of cleaner production technologies. The talent introduction policy should be improved, in addition to the overall human capital level of the region, and the management ability of the water resource system should be strengthened. Environmental regulation policies should be implemented and enterprises given appropriate tax incentives and reduced production costs, avoiding the negative impacts on the coupling coordinated development of urban and rural water poverty.

## 5. Limitations

This paper has the following shortcomings: First, these recommendations are based on research results and the results of spatial convergence models. The recommendations are more specific to the current water management in the northwest territories; therefore, it is hard to replicate with 100 percent accuracy in other regions. The study of the analytical results of any given region may not result in fully applicable policy recommendations. However, it is believed that this analytical paradigm has a strong reference value for other regions. Using these paradigms to analyze the water situation in other regions, policy recommendations can be more realistic. Second, the selection criteria and weighting method for the different indicators would likely produce different wp results. In the future, there is still a need to use a variety of other indicators, evaluation standards and methods; an analysis of the existing robustness; and a reliability of inspection. Third, in addition to the spatial hot spot analysis and spatial autocorrelation analysis, the future research focus of the authors of this paper will be to use spatial convergence, Durbin, hysteresis and other models to explore the influencing factors from the perspective of spatio-temporal factors. Fourthly, considering the availability and operability of the data, city–level administrative units in northwest China were selected as the research objects. If more micro-level and specific regions (prefecture- or county-level) were selected in the future, the spatial-temporal characteristics of water poverty could be more accurately reflected. Fifth, because of the different physical and geographical conditions, there were significant regional differences. In the future, specific zoning research can be conducted in combination with the different development goals of each region. 

## 6. Conclusions

Based on the “pain point” of the water resource system in the context of the “urban–rural dual structure” in northwest China, this paper focused on the natural characteristics and practical problems of the study area and combined the WPI model to construct the evaluation index system of urban water poverty and rural water poverty. It measured the coupling coordinated development level of urban and rural water resource systems in 52 regions in northwest China from 2000 to 2020 and studied the spatial-temporal differentiation. Further, the random effect panel Tobit model was used to analyze its driving factors. The main results were as follows.

Firstly, from 2000 to 2020, the scores of urban water poverty showed an upward trend, the development speed of urban water resources gradually slowed down and water resources as a whole developed in a balanced direction, whereas the scores of rural water poverty increased significantly. The improvement speed of rural water resources has been accelerating, but the two-level differentiation between regions has gradually intensified. In general, the improvement speed of urban water resources is lagging behind that of rural water resources, and the gap between urban and rural water resource development is gradually narrowing. Spatially, the high-value area of urban water poverty is gradually moving eastward, while the low-value area is gradually moving westward. 

Secondly, from the perspective of the coupling coordinated development of urban and rural water poverty, in 2020, nearly 70% of the regions were still basically uncoordinated, and the overall situation is not optimistic. Spatially, the coupling coordinated development of urban and rural water poverty shows obvious characteristics of spatial agglomeration. The agglomeration first increased and then decreased, and the heterogeneity first decreased and then increased. The level of urban–rural water poverty coupling coordination changed significantly in cold and hot spots, and the spatial evolution path was dominated by successive transfer and supplemented by cross-level transfer. The hot spot area in the southeast is gradually expanding, and the cold spot area in the middle is gradually shrinking.

Thirdly, from the perspective of the driving factors of the coupling coordinated development of urban and rural water poverty, at the benchmark regression level, the level of economic development, industrial structure, agricultural production level and agricultural expenditure had a positive impact on the coupling coordinated development of urban and rural water poverty in northwest China. Furthermore, the degree of industrialization and labor transfer had a negative impact on the coupling coordinated development of urban and rural water poverty in northwest China. At the level of regional heterogeneity, the influencing factors varied greatly.

## Figures and Tables

**Figure 1 ijerph-20-02043-f001:**
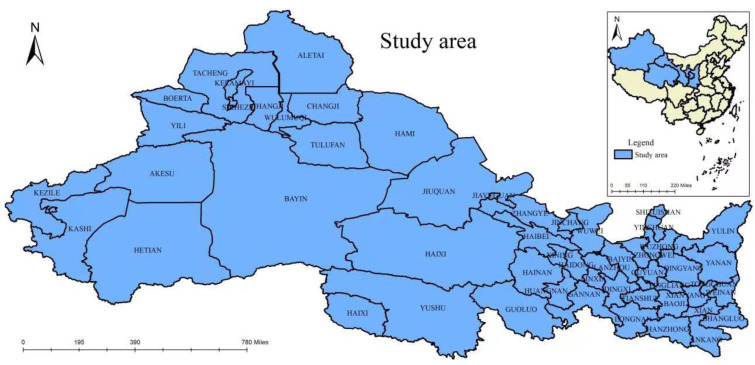
Study area.

**Figure 2 ijerph-20-02043-f002:**
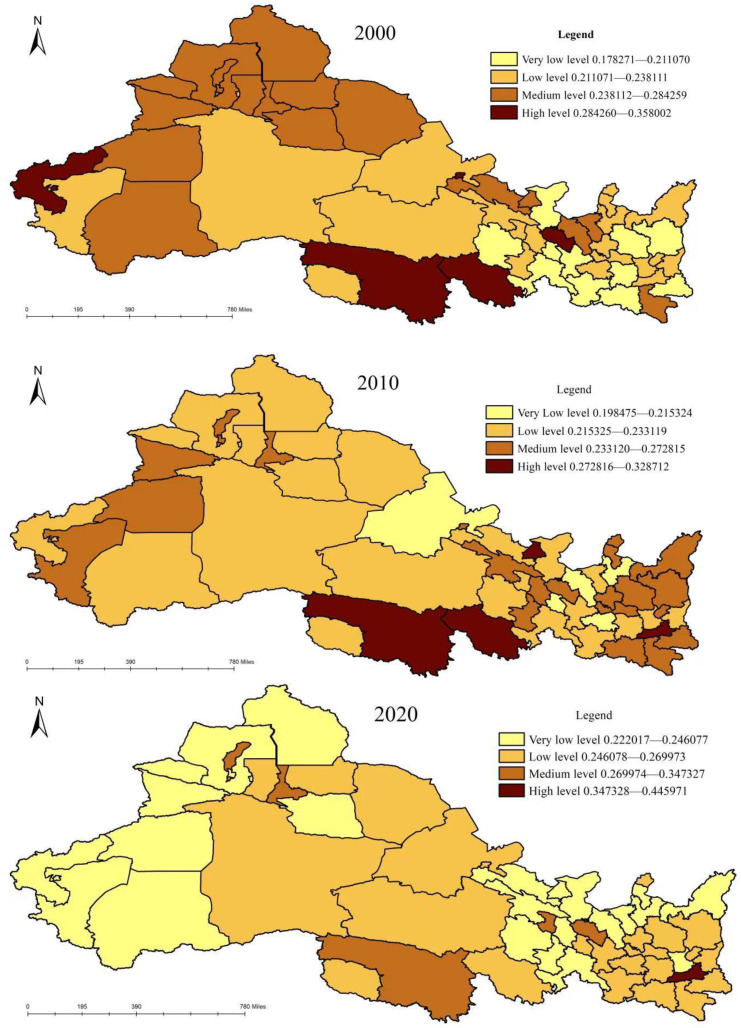
Urban water poverty level in northwest China from 2000 to 2020.

**Figure 3 ijerph-20-02043-f003:**
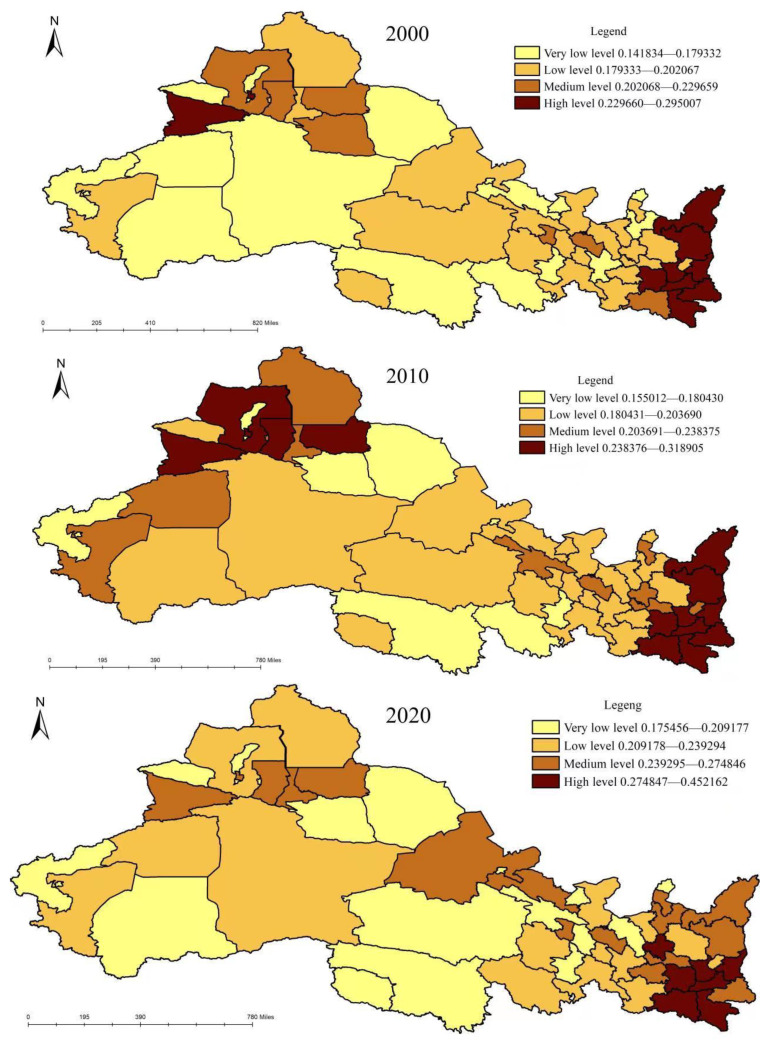
Rural water poverty level in northwest China from 2000 to 2020.

**Figure 4 ijerph-20-02043-f004:**
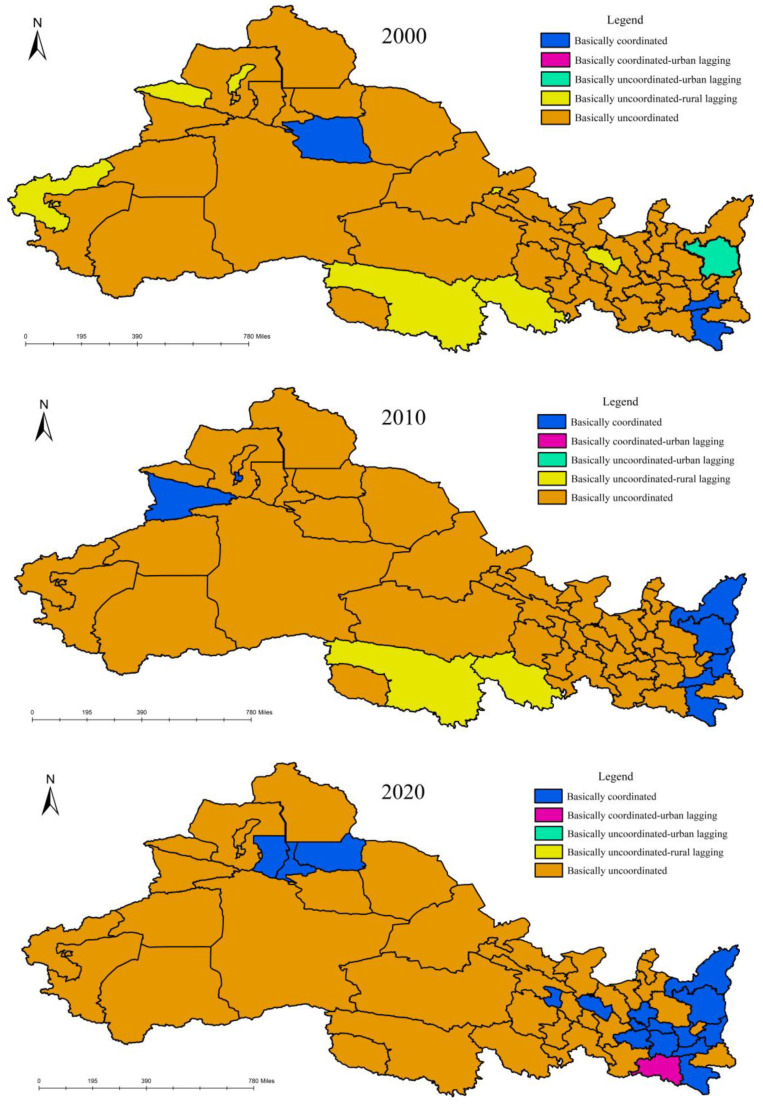
Coupling coordination level between urban and rural water poverty in northwest China from 2000 to 2020.

**Table 1 ijerph-20-02043-t001:** Weights of the urban and rural WPI components, variables.

System	Component	Variable	AHP	PCA	Integrated
**Urban**	Resource (0.2)	Per capita annual urban water resources (m^3^) [14]	0.1667	0.0689	0.1178
		Variation of rainfall (mm) [14]	0.0333	0.0615	0.0474
	Access (0.2)	Length of water supply pipeline (km) [15]	0.0343	0.1015	0.0679
		Length of drainage pipeline (km) [23]	0.1501	0.1030	0.1265
		Urban tap water popularity rate (%) [24]	0.0156	0.0765	0.0461
	Capacity (0.2)	Government financial self-sufficiency rate (%) [14]	0.1461	0.0968	0.1214
		Higher education popularity rate (%) [19]	0.0162	0.0533	0.0347
		Per capita urban disposable income (CNY) [23]	0.0377	0.0660	0.0518
	Use (0.2)	Urban per capita domestic water consumption (L) [15]	0.1800	0.0801	0.1300
		Industrial water use per CNY 10,000 (m^3^) [14]	0.0200	0.0562	0.0381
	Environment (0.2)	Volume of industrial wastewater (m^3^) [24]	0.0704	0.0892	0.0798
		Per capita urban green space area (m^3^) [19]	0.0177	0.0914	0.0546
		Sewage treatment efficiency (%) [15]	0.1118	0.0557	0.0838
**Rural**	Resource (0.2)	Per capita annual rural water resources (m^3^) [14]	0.0250	0.0327	0.0289
		Variation of rainfall (mm) [26]	0.1750	0.0820	0.1285
	Access (0.2)	The actual irrigation area (m^2^) [19]	0.0343	0.0861	0.0602
		Number of reservoirs [26]	0.1501	0.0820	0.1160
		Percentage of agricultural water (%) [23]	0.0156	0.0551	0.0354
	Capacity (0.2)	Number of doctors per 10,000 people () [15]	0.1528	0.1291	0.1409
		Primary education popularity rate (%) [19]	0.0230	0.0375	0.0303
		Per capita rural disposable income (CNY) [15]	0.0242	0.1041	0.0642
	Use (0.2)	Rural per capita domestic water consumption (L) [23]	0.0250	0.0456	0.0353
		Agriculture water use per CNY 10,000 (m^3^) [14]	0.1750	0.0673	0.1212
	Environment (0.2)	Chemical fertilizer use per hectare (t/hm^2^) [24]	0.1348	0.0739	0.1044
		Numbers of toilets per 10,000 people [23]	0.0201	0.1198	0.0699
		Soil and water loss control area (km^2^) [19]	0.0451	0.0848	0.0650

**Table 2 ijerph-20-02043-t002:** The coupling coordination types of urban and rural water poverty.

Categories	Value Range	Subcategories	Value Range	Coupling Coordination Types
Coordinated development	0.8 < *D* ≤ 1	Advanced coordination	*U*_1_–*U*_2_ > 0.1	Advanced coordination-rural lagging type
			*U*_2_–*U*_1_ > 0.1	Advanced coordination-urban lagging type
			0 < |*U*_1_–*U*_2_| < 0.1	Advanced coordination
Transformation development	0.5 < *D* ≤ 0.8	Basically coordinated	*U*_1_–*U*_2_ > 0.1	Basically coordinated-rural lagging type
			*U*_2_–*U*_1_ > 0.1	Basically coordinated-urban lagging type
			0 < |*U*_1_–*U*_2_| < 0.1	Basically coordinated
Uncoordinated development	0.3 < *D* ≤ 0.5	Basically uncoordinated	*U*_1_–*U*_2_ > 0.1	Basically uncoordinated-rural lagging type
			*U*_2_–*U*_1_ > 0.1	Basically uncoordinated-urban lagging type
			0 < |*U*_1_–*U*_2_| < 0.1	Basically uncoordinated
	0 < *D* ≤ 0.3	Seriously uncoordinated	*U*_1_–*U*_2_ > 0.1	Seriously uncoordinated-rural lagging type
			*U*_2_–*U*_1_ > 0.1	Seriously uncoordinated-urban lagging type
			0 < |*U*_1_–*U*_2_| < 0.1	Seriously uncoordinated

**Table 3 ijerph-20-02043-t003:** The driving factors of coupling coordination degree.

	Variable	Abbreviation	Description
Dependent variable	the degree of coupling coordination [32]	*D*	Calculation results of coupling coordination degree model
Independent variable	the degree of industrialization [33]	*indu*	Percentage of secondary industry in GDP (%)
	the level of opening-up [33]	*open*	Annual actual utilized foreign direct investment (CNY 10,000)
	technological progress [34]	*tech*	Research and experimental development (R&D) expenditure (CNY 10,000)
	population size [34]	*per*	Number of registered residence population
	the level of economic development [38]	*pgdp*	Per capita gross domestic product (CNY)
	industrial structure [29]	*is*	Percentage of tertiary industry in GDP (%)
	labor transfer [31]	*labtra*	Percentage of non-agricultural population in the total population (%)
	agricultural expenditure [29]	*fina*	Agricultural, forestry and water expenditure in local fiscal expenditure (CNY 10,000)
	the level of agricultural production [31]	*pgrain*	Per capita grain output (t)
	environmental regulation [29]	*er*	Industrial output/volume of industrial wastewater (10,000 CNY/m^3^)

**Table 4 ijerph-20-02043-t004:** Cold and hot spot results of coordinated development.

Year	Hot Spots	Cold Spots
2000	Xian, Tongchuan, Baoji, Xianyang, Weinan, Yanan, Hanzhong, Ankang, Shangluo, Qingyang	Lanzhou, Baiyin, Wuwei, Zhangye, Jiuquan, Linxia, Haidong, Haibei
2010	Xian, Tongchuan, Baoji, Xianyang, Weinan, Yanan, Hanzhong, Ankang, Shangluo, Qingyang	Lanzhou, Jinchang, Baiyin, Wuwei, Zhangye, Jiuquan, Linxia, Gannan, Xining, Haidong, Haibei, Huangnan, Hami, Bayangol
2020	Xian, Tongchuan, Baoji, Xianyang, Weinan, Yanan, Hanzhong, Ankang, Shangluo, Tianshui, Pingliang, Qingyang, Longnan, Guyuan	Jiayuguan, Jinchang, Zhangye, Xining, Haibei, Bayangol, Kizilsu, Kashgar

**Table 5 ijerph-20-02043-t005:** Regression results.

Variables	Total		Shaanxi		Gansu	
	Coefficients	*p*-Value	Coefficients	*p*-Value	Coefficients	*p*-Value
indu	0.0005	0.5260	−0.0053	0.5350	0.0049	0.0830 *
open	−0.0281	0.2650	0.0072	0.0190 **	−0.0592	0.0420 **
Tech	0.0144	0.0000 ***	−0.0005	0.9000	0.0238	0.0000 ***
Per	0.0001	0.8400	1.9116	0.0740 *	1.0456	0.3940
pgdp	0.0082	0.0000 ***	0.0278	0.0000 ***	0.0083	0.0000 ***
Is	0.0049	0.0000 ***	−0.0030	0.6640	0.0045	0.0560 *
labtra	−0.0011	0.1930	0.0018	0.6560	−0.0006	0.8020
Fina	0.0091	0.0000 ***	−0.0049	0.0180 **	0.0103	0.0000 ***
pgrain	0.0026	0.0050 ***	−0.0054	0.5340	0.0086	0.0010 ***
Er	−0.0001	0.8790	0.0166	0.0000 ***	0.0064	0.3010
cons	0.4711	0.0000 ***	0.5560	0.0000 ***	0.4905	0.0000 ***
**Variables**	**Ningxia**		**Qinghai**		**Xinjiang**	
	**Coefficients**	***p*-Value**	**Coefficients**	***p*-Value**	**Coefficients**	***p*-Value**
indu	−0.0094	0.0000 ***	−0.0002	0.8690	−0.0061	0.0160 **
open	−0.0325	0.0740 *	−0.0184	0.3600	0.0188	0.1000 *
Tech	−0.0491	0.0150 **	0.0320	0.2180	0.0174	0.0450 **
Per	−0.0002	0.9510	−3.2386	0.0450 **	3.1633	0.0000 ***
pgdp	0.0077	0.0530 *	0.0081	0.0460 **	0.0084	0.0000 ***
Is	0.0023	0.0690 *	0.0058	0.0230 **	0.0001	0.9250
labtra	0.0033	0.2300	−0.0128	0.0040 ***	−0.0004	0.9760
Fina	0.0418	0.0000 ***	0.0196	0.0030 ***	0.0087	0.0000 ***
pgrain	0.0152	0.0150 **	0.0202	0.0210 **	0.0023	0.0160 **
Er	0.0001	0.9670	0.0048	0.4330	−0.0154	0.0070 ***
cons	0.4749	0.0000 ***	0.3781	0.0000 ***	0.5641	0.0000 ***

*, **, *** mean significant at the level of 10%, 5% and 1%.

## Data Availability

Data are available on request due to restrictions, e.g., privacy or ethical restrictions. The data presented in this study are available on request from the corresponding author. The data are not publicly available due to the strict management of various data and technical resources within the research teams.

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
