# Peer review of "The Coupling Coordination Measurement, Spatio-Temporal Differentiation and Driving Mechanism of Urban and Rural Water Poverty in Northwest China"

_ijerph, 2023, doi:10.3390/ijerph20032043_

Round 1
Reviewer 1 Report
The Article „Research on the coupling coordination measurement, spatio-temporal differentiation and driving mechanism of urban and rural water poverty in Northwest China” refers to the relationship between urban and rural water resources systems is of great significance for alleviating water use contradictions and optimizing water resources allocation. The Authors quantified the relationship between the urban and the rural water resources system in Northwest China from 2000 to 2020, and further, the spatial and temporal evolution characteristics and driving mechanism were studied by using spatial autocorrelation, hot-spot analysis and Tobit model.
The article is in line with current scientific trends. The results obtained by the authors and the conclusions reached from the research can be evaluated as a new contribution to scientific knowledge in the analyzed topic.
The article is methodically well written. It contains all the elements required in a good scientific article. In terms of content, I rate the article well.
I have minor comments, the inclusion of which should further improve the quality of the article.
1 I would suggest part of the content from Chapters 3 and 4 be separated as a separate chapter: Discussion. Chapters 3 and 4 can be combined into one, and I suggest leaving only the research results themselves in the chapter without their discussion.
2 I suggest in Chapter 2 to add a subsection: study area (in it a map with all the names - then there will be no need to repeat this element in Figure 1). Also, please clearly describe in this chapter the sources and types of data used in the analyses.
After making these above minor corrections, I recommend the article for publication.
Author Response
Responses to reviewers 1
AE1: I would suggest part of the content from Chapters 3 and 4 be separated as a separate chapter: Discussion. Chapters 3 and 4 can be combined into one, and I suggest leaving only the research results themselves in the chapter without their discussion.
Response: Thank you for this helpful suggestion. We thank your comments which helped us improve the quality of this manuscript. Firstly, I have combined Chapters 3 and 4. Secondly, I have moved 4.1 (Selection of variables) to 2.5. Thirdly, I have moved line 308-309, line 328-330, line 346-348, line 387-396, line 417-419, line 535-542 to Chapters 4 (Conclusions and Discussion).
AE2: I suggest in Chapter 2 to add a subsection: study area (in it a map with all the names - then there will be no need to repeat this element in Figure 1). Also, please clearly describe in this chapter the sources and types of data used in the analyses.
Response: Thank you for this helpful suggestion. We thank your comments which helped us improve the quality of this manuscript. Firstly, I have added study area. It is that “Northwest China belongs to the dry early morning and semi-dry early morning region of China, which mainly includes Xinjiang, Ningxia, Qinghai, Gansu, Shaanxi and other provinces. The land area accounts for one-third of the total area of China, and the population accounts for approximately 7% of the total population of the country. However, the water resources only account for approximate 4% of the total water re-sources of the country, and the per capita water resources only account for 80% of the per capita level of the country. With the growth in population and economic develop-ment, the water consumption and the demand for water increases, and the exploitation and utilization of water resources gradually increase. At the same time, there is a serious waste of water resources. Due to natural and human-made reasons, the contradiction between the supply and demand of water resources is more prominent, resulting in serious shortage of ecological, industrial, agricultural and urban water, resulting in dry water surfaces, shrinkage of the oasis, riverbed flow interruption, water pollution and deterioration of the ecological environment deterioration in Northwest China. At pre-sent, the ecological environment in Northwest China is extremely fragile due to drought and water shortage. However, the people's exploitation of water resources for produc-tion, living and survival has not decreased, which has greatly exceeded the carrying capacity of water resources in the region. In addition, people's awareness of ecological environment protection is not strong. As a result, the region is experiencing a water resource shortage, production development is blocked and water for economic devel-opment is crowding out the ecological water situation. The water resource shortage has become the restrictive factor of the development and sustainable development of Northwest China.
Firstly, Northwest China is located inland, far away from the ocean and cut off by mountains. This region experiences little precipitation, and the limited precipitation is mainly concentrated in the summer and autumn; the summer water shortage is very serious. Not only do water resources fail to meet the needs of agricultural irrigation and industrial production during a shortage, but people in many places even have trouble reserving water. In Northwest China, there are only 30 large reservoirs, which have a low regulation of water volume and engineering water shortage, resulting in severe agricultural spring drought and a greatly reduced grain output; this even means that grain self-sufficiency is not possible. For example, at present, diversion projects with a low control capacity are still the main means of water supply in Xinjiang, and the water supply capacity of regulatory reservoir projects only accounts for 14.4% of the total water supply capacity; the plain reservoirs are the main reservoirs, while the controlled reservoirs in mountainous areas are very low in scale, resulting in a situation of low drought resistance ability and poor spatial and temporal regulation ability. The spring irrigation period results in a serious water shortage, and the flood season irrigation, not only results in waste water, but also aggravates the secondary salinization of land. As another example, in the existing water supply projects in Shaanxi, there are few water storage projects. On large rivers in the Guanzhong area, where water shortage is serious, three quarters of farmland irrigation can be supplied by diversion projects, and the de-gree of water supply assurance is very low.
Secondly, the spatial and temporal distribution of water resources in Northwest China is more uneven, which means it is difficult touse effectively. At the same time, as the temperature rises, glaciers and mountain snow melt away, lakes and wetlands shrink, evaporation increases, and available water sources in different regions basically decrease further. The surface runoff is mainly concentrated in the flood season, ac-counting for more than 60% of the annual runoff. It is precisely because of the influence of this precipitation characteristic that the utilization of water resources is greatly re-stricted and disasters are frequent, resulting in the instability of industrial and agri-cultural production. The northwest region has a lack of water resources, and most of the regions west of Lanzhou, namely, Ri, Ning, Inner Mongolia, Qing and Xin, have an annual precipitation of less than 300 mm. For more than 1 million square kilometers of desert and the Gobi area, annual rainfall is as low as 50 mm or less. The air is dry and the evaporation is vigorous. Drought is defined when the dryness index (the ratio of evaporation capacity to precipitation) is greater than 1. In northwest China, it can be greater than 50. Water and life have the same significance. In the past 20 years, the annual rainfall has increased in some parts of Northwest China, but the spatio-temporal distribution has been more uniform.
Thirdly, the water efficiency of industry and agriculture in Northwest China is low. Industrial water technology is backward and the reuse rate of water is very low. For example, the total water consumption per ten thousand yuan of output value of the main industries in Lanzhou, Gansu is 370t, Urumqi, Xinjiang is 707t, and Xining, Qinghai is 1764t, which are 3.8 times, 7.3 times and 18 times that of water-saving Qingdao, Shandong (97t), respectively. Agricultural water waste is even more shocking, ac-counting for more than 70% to 90% of the total water consumption. Some old irrigation areas have, for a long time, used flood irrigation and string irrigation; however, field projects do not match and canal system leakage is serious; the effective benefit coeffi-cient of the canal system is generally 0.4, and the highest is 0.54. In terms of the exten-sive agricultural management, the irrigation quota is too high.” in line 341-415. Secondly, I have revised Figure 1.
Figure 1. Study area
Thirdly, about data, I have revised Table 1.
Table 1. Weights of the urban and rural WPI components, variables.
|
System |
Component |
Variable |
AHP |
PCA |
Integrated |
|
Urban |
Resource (0.2) |
Per capita annual urban water resources (m3) [14] |
0.1667 |
0.0689 |
0.1178 |
|
|
Variation of rainfall(mm)[14] |
0.0333 |
0.0615 |
0.0474 |
|
|
Access (0.2) |
Length of water supply pipeline (km) [15] |
0.0343 |
0.1015 |
0.0679 |
|
|
|
Length of drainage pipeline(km) [23] |
0.1501 |
0.1030 |
0.1265 |
|
|
|
Urban tap water popularity rate(%) [24] |
0.0156 |
0.0765 |
0.0461 |
|
|
Capacity (0.2) |
Government financial self-sufficiency rate(%)[14] |
0.1461 |
0.0968 |
0.1214 |
|
|
|
Higher Education popularity rate(%) [19] |
0.0162 |
0.0533 |
0.0347 |
|
|
|
Per capita urban disposable income(yuan) [23] |
0.0377 |
0.0660 |
0.0518 |
|
|
Use (0.2) |
Urban per capita domestic water consumption(L) [15] |
0.1800 |
0.0801 |
0.1300 |
|
|
|
Industrial water use per 10,000 yuan(m3) [14] |
0.0200 |
0.0562 |
0.0381 |
|
|
Environment (0.2) |
Volume of industrial wastewater(m3) [24] |
0.0704 |
0.0892 |
0.0798 |
|
|
|
Per capita urban green space area(m3) [19] |
0.0177 |
0.0914 |
0.0546 |
|
|
|
Sewage treatment efficiency(%) [15] |
0.1118 |
0.0557 |
0.0838 |
|
|
Rural |
Resource (0.2) |
Per capita annual rural water resources (m3) [14] |
0.0250 |
0.0327 |
0.0289 |
|
|
Variation of rainfall(mm) [26] |
0.1750 |
0.0820 |
0.1285 |
|
|
Access (0.2) |
The actual irrigation area (m2) [19] |
0.0343 |
0.0861 |
0.0602 |
|
|
|
Numbers of reservoir [26] |
0.1501 |
0.0820 |
0.1160 |
|
|
|
Percentage of agricultural water(%)[23] |
0.0156 |
0.0551 |
0.0354 |
|
|
Capacity (0.2) |
Number of doctors per 10,000 people () [15] |
0.1528 |
0.1291 |
0.1409 |
|
|
|
Primary Education popularity rate(%) [19] |
0.0230 |
0.0375 |
0.0303 |
|
|
|
Per capita rural disposable income(yuan) [15] |
0.0242 |
0.1041 |
0.0642 |
|
|
Use (0.2) |
Rural per capita domestic water consumption(L) [23] |
0.0250 |
0.0456 |
0.0353 |
|
|
|
Agriculture water use per 10,000 yuan(m3) [14] |
0.1750 |
0.0673 |
0.1212 |
|
|
Environment (0.2) |
Chemical fertilizer use per hectare(t/hm2) [24] |
0.1348 |
0.0739 |
0.1044 |
|
|
|
Numbers of toilets per 10,000 people [23] |
0.0201 |
0.1198 |
0.0699 |
|
|
|
Soil and water loss control area (km2) [19] |
0.0451 |
0.0848 |
0.0650 |

Reviewer 2 Report
It is good work and interesting. Please answer the following questions
Minor comments
equation 1.
Are there others reported to do the same calculation?
equation 4.
Are there others reported to do the same calculation?
Major comments
Based on the findings of the above research, this paper proposes the following urban and rural water resources management policies:
this could be replicated for other areas in China and the world?
Author Response
Responses to reviewers 2
AE1: equation 1. Are there others reported to do the same calculation?
Response: Thank you for this helpful suggestion. We thank your comments which helped us improve the quality of this manuscript. Equation 1 had been used in many literatures. And I have added quotes. It is that “where, wi refers to weight, resource, access, capacity, use, and environment represent the weighted scores of the indicators of five dimensions after data standardization; WPI is the score of water poverty. The lower the value, the more serious the shortage of water resources [19, 21].” in line 217-220.
AE2: equation 4. Are there others reported to do the same calculation?
Response: Thank you for this helpful suggestion. We thank your comments which helped us improve the quality of this manuscript. Equation 4 had been used in many literatures. And I have added quotes. It is that “Spatial autocorrelation is a common method used to analyze the spatial heteroge-neity and correlation of data, which is are used to explain the correlation characteristics of the spatial attribute data. The global Moran’s I index is was used to measure the spatial correlation characteristics of urban–-rural water poverty coupling coordination degree in Northwest China, and its spatial agglomeration situation is analyzed [20].” in line 301-305.
AE3: Based on the findings of the above research, this paper proposes the following urban and rural water resources management policies: this could be replicated for other areas in China and the world?
Response: Thank you for this helpful suggestion. We thank your comments which helped us improve the quality of this manuscript.
These recommendations are based on research results and the results of spatial convergence models. The recommendations are more specific to current water management in the Northwest Territories, so I think it's hard to replicate 100 percent in other regions. The study of the analytical results of any given region may not give policy recommendations on these results full applicability. However, I believe that this analytical paradigm has strong reference value for other regions. Using these paradigms to analyze the water situation in other regions, policy recommendations can be more realistic.

Reviewer 3 Report
1. It is suggested to delete "Research on" in the title as it is not a meaningful title.
2. In the abstract, the author declared that the scores of urban water poverty and rural water poverty have risen significantly. This is confused because I think the coupling coordination model can not result in improvement of the score of proverty.
3. The author should focus on the shortcomings of the existing research in the introduction and analyze it, pointing out the research gap.
4.The word count of conclusion is too much, and conclusion should focus on the main findings of the study. Add a discussion section, and put the main content of conclusion in the discussion section.
5.It is recommended that the authors conclude by pointing out the shortcomings of this study and giving future research perspectives
6.There are many grammatical errors in the text and it is recommended to invite native English speakers to revise the language
Author Response
Responses to reviewers 3
AE1: It is suggested to delete "Research on" in the title as it is not a meaningful title.
Response: Thank you for this helpful suggestion. We thank your comments which helped us improve the quality of this manuscript. I have deleted it. The new title is “The coupling coordination measurement, spatio-temporal differentiation and driving mechanism of urban and rural water poverty in Northwest China”.
AE2: In the abstract, the author declared that the scores of urban water poverty and rural water poverty have risen significantly. This is confused because I think the coupling coordination model can not result in improvement of the score of poverty.
Response: Thank you for this helpful suggestion. We thank your comments which helped us improve the quality of this manuscript. It is not concluded that coupling coordination will lead to the increase of urban and rural water poverty scores. Conversely, an increase in urban and rural water poverty scores may lead to an increase in coupling coordination. This is because the empirical analysis in this paper is mainly data-driven, and the coupling coordination degree is calculated based on urban and rural water poverty scores. Then the improvement of urban and rural water resources may have a positive impact on the coupling and coordination of urban-rural water resources.
AE3: The author should focus on the shortcomings of the existing research in the introduction and analyze it, pointing out the research gap.
Response: Thank you for this helpful suggestion. We thank your comments which helped us improve the quality of this manuscript. I have revised it. It is that “The above research results have revealed the formation mechanism, and temporal and spatial characteristics of water poverty through the superposition of quantitative mod-els, expanded the theoretical evaluation method of water poverty, and provided a good research foundation for this paper. Nevertheless, there are still many problems to be solved in the study of water poverty. Firstly, the index system of the water poverty measurement model is not reasonable. On the basis of pursuing comprehensiveness, a large number of indicators are used to evaluate the water resource system. The selection of indicators is one-sided and limited, which initially outlines the multi-dimensional characteristics of the water resource system. However, it focuses more on the description of resource endowment, and pays insufficient attention to human welfare or the social development level, which cannot comprehensively interpret the rich connotation of water resources sustainability. Based on the perspective of poverty concept, "promoting common prosperity through high-quality development" undoubtedly adds a new meaning to the connotation of poverty–human well-being. Secondly, there have been few studies on the interaction mechanism between urban and rural water re-sources. The urban–rural dual structure left over from China's planned economy era determines the dominant role of cities in water resource utilization and solidifies the unbalanced pattern of water resource distribution between urban and rural areas. Attention should be paid to the temporal and spatial characteristics of urban and rural water resources, but this problem has not been paid enough attention. Most studies on the dynamic relationship between urban and rural water resources have focused on the qualitative research level, and no studies have directly measured the coupling coordination level of urban and rural water resources, the historical context, or the lack of attention that has been paid to regional differences. The Chinese government has pointed out that "on the basis of continuing to promote development, efforts should be made to solve the problem of unbalanced and inadequate development". Therefore, it is necessary to further grasp the regional differences, and sources of the coupling and coordination of water resources between urban and rural areas in China, comprehensively analyze the current situation and underlying causes of regional unbalanced development, provide a factual basis for promoting regional coupling and coordination development, and provide reference ideas for solving the problems of unbalanced and in-adequate development. Finally, existing studies lack a reasonable explanation of the driving factors of the interaction between urban and rural water resources. Although China has introduced relevant water resource management policies and obtained successful cases, as presented by the authors in [22], there is still a lack of practical experience in improving urban and rural water resource systems in different regions, resulting in a persistent contradiction between urban and rural water resources allocation. Studies on the influencing factors of urban and rural water resources are limited to a certain degree in urban or rural areas, and lack investigation into the overall impact of urban and rural water resources. Therefore, in-depth discussion on the mechanism of influencing factors is needed to further investigate the possible regional heterogeneity of influencing factors.” in line 115-150.
AE4: The word count of conclusion is too much, and conclusion should focus on the main findings of the study. Add a discussion section, and put the main content of conclusion in the discussion section.
Response: Thank you for this helpful suggestion. We thank your comments which helped us improve the quality of this manuscript. I have revised the “conclusion”. It is that “Based on the “pain point” of the water resource system in the context of the “urban–rural dual structure” in Northwest China, this paper focused on the natural characteristics and practical problems of the study area, and combined WPI model to construct the evaluation index system of urban water poverty and rural water poverty. It measured the coupling coordinated development level of urban and rural water resource systems in 52 regions in Northwest China from 2000 to 2020 and studied the spatial-temporal differentiation. Further, the random effect panel Tobit model was used to analyze its driving factors. The main results were as follows:
Firstly, from 2000 to 2020, the scores of urban water poverty showed an upward trend, the development speed of urban water resources gradually slowed down, and water resources as a whole developed in a balanced direction, whereas the scores of rural water poverty increased significantly. The improvement speed of rural water resources has been accelerating, but the two-level differentiation between regions has gradually intensified. In general, the improvement speed of urban water resources is lagging behind that of rural water resources, and the gap between urban and rural water resources development is gradually narrowing. Spatially, the high-value area of urban water poverty is gradually moving eastward, while the low-value area is gradually moving westward.
Secondly, from the perspective of the coupling coordinated development of urban and rural water poverty, in 2020, nearly 70% of the regions were still basically uncoordinated, and the overall situation is not optimistic. Spatially, the coupling coordinated development of urban and rural water poverty shows obvious characteristics of spatial agglomeration. The agglomeration first increased and then decreased, and the heterogeneity first decreased and then increased. The level of urban–rural water poverty coupling coordination changed significantly in cold and hot spots, and the spatial evolution path was dominated by successive transfer, supplemented by cross level transfer. The hot-spot area in the southeast is gradually expanding, and the cold spot area in the middle is gradually shrinking.
Thirdly, from the perspective of the driving factors of the coupling coordinated development of urban and rural water poverty, at the benchmark regression level, the level of economic development, industrial structure, agricultural production level and agricultural expenditure had a positive impact on the coupling coordinated development of urban and rural water poverty in Northwest China, and the degree of industrialization and labor transfer had a negative impact on the coupling coordinated development of urban and rural water poverty in Northwest China. At the level of regional heterogeneity, the influencing factors varied greatly. ” in line 816-874. In addition, I added the “discussion” in line 925-946.
AE5: It is recommended that the authors conclude by pointing out the shortcomings of this study and giving future research perspectives.
Response: Thank you for this helpful suggestion. We thank your comments which helped us improve the quality of this manuscript. I have revised it. It is that “This paper has the following shortcomings: First, these recommendations are based on research results and the results of spatial convergence models. The recommendations are more specific to the current water management in the northwest territories; therefore, it is hard to replicate with 100 percent accuracy in other regions. The study of the analytical results of any given region may not result in fully applicable policy recommendations. However, it is believed that, this ana-lytical paradigm has a strong reference value for other regions. Using these paradigms to analyze the water situation in other regions, policy recommendations can be more realistic. Second, the selection criteria and weighting method for the different indicators would likely produce different wp results. In the future, there is still a need to use a variety of other indicators, evaluation standards and methods, analysis of the existing robustness, and a reliability of inspection. Third, in addition to spatial hotspot analysis and spatial autocorrelation analysis, the future research focus of the authors of this paper will be to use spatial convergence, Durbin, hysteresis and other models to explore the influencing factors from the perspective of spatio-temporal factors. Fourthly, considering the availability and operability of the data, city–level administrative units in North-west China were selected as the research objects. If more micro-level and specific regions (prefecture- or county-level) were selected in the future, the spatial-temporal characteristics of water poverty could be more accurately reflected. Fifth, because of the different physical and geo-graphical conditions, there were significant regional differences. In the future, specific zoning research can be conducted in combination with the different development goals of each region.” in line 925-946.
AE6: There are many grammatical errors in the text and it is recommended to invite native English speakers to revise the language
Response: Thank you for this helpful suggestion. We thank your comments which helped us improve the quality of this manuscript. I have use English editing from MDPI to revised it. It is that english-edited-57868
